# Effects of Substitution of Higher-Alcohol Products with Lower-Alcohol Products on Population-Level Alcohol Purchases: ARIMA Analyses of Spanish Household Data

**DOI:** 10.3390/nu14194209

**Published:** 2022-10-10

**Authors:** Peter Anderson, Daša Kokole

**Affiliations:** 1Faculty of Health, Medicine and Life Sciences, Maastricht University, 6200 MD Maastricht, The Netherlands; 2Population Health Sciences Institute, Newcastle University, Baddiley-Clark Building, Newcastle upon Tyne NE2 4AX, UK

**Keywords:** no-alcohol products, product reformulation, household purchase data, Spain, l, ARIMA modelling

## Abstract

In its action plan (2022–2030) to reduce the harmful use of alcohol, the WHO calls on economic operators to “substitute, whenever possible, higher-alcohol products with no-alcohol and lower-alcohol products in their overall product portfolios, with the goal of decreasing the overall levels of alcohol consumption in populations and consumer groups”. This paper investigates substitution at the level of the consumer, based on Spanish household purchase data. ARIMA modelling of market research data of 1.29 million alcohol purchases from 18,954 Spanish households is used to study the potential impact of lower-strength alcohol products and the impact of beer prices in reducing household purchases of grams of alcohol between the 2nd quarter of 2017 and 1st quarter of 2022. Reducing the alcohol strength of existing higher-strength beers and wines had a much greater associated impact on reducing the purchases of all grams of alcohol than the relatively small increases in purchases of no-alcohol beers (ABV ≤ 1.0%) and zero-alcohol wines (ABV = 0.0%). For beers, the relative price per gram of alcohol decreased with the increasing ABV of the beer. Increasing the price per gram of alcohol in beers with an ABV > 3.5%, adjusted for the ABV of the beer, was associated with much greater increases in purchases of no-alcohol beers (ABV ≤ 1.0%) and much greater decreases in purchases of all grams of alcohol than decreases in the price of no-alcohol beers or increases in the price of beers with an ABV > 3.5% unadjusted for ABV. Thus, a key to reducing purchases of grams of alcohol, which also results in increased purchases of no-alcohol beers, is to increase the price of higher strength beers (ABV > 3.5%) with the price per gram of alcohol increasing as the ABV of the product increases.

## 1. Introduction

The risk of ill-health and premature death increases with increasing levels of alcohol consumption from a minimum exposure level of about five grams of ethanol (half a drink) per day, with the risk greater for younger, as opposed to older drinkers [1]. To reduce the harm caused by alcohol, consumers need to drink less alcohol [1]. The WHO, in its draft action plan (2022–2030) to effectively implement the global strategy to reduce the harmful use of alcohol as a public health priority, in addition to the “continued enforcement of high-impact cost-effective policy options included in its SAFER technical package” [2], called on ‘economic operators’ to “substitute, whenever possible, higher-alcohol products with no-alcohol and lower-alcohol products in their overall product portfolios, with the goal of decreasing the overall levels of alcohol consumption in populations and consumer groups” [3].

The WHO’s call on economic operators is based on an assumption that consumers substitute higher-alcohol products with no-alcohol and lower-alcohol products and thus buy and drink less grams of alcohol [4]. Elsewhere, based on British [5,6] and Spanish [7] household purchase data, we have demonstrated that, in general, consumers do seem to substitute higher-strength alcohol products with lower-strength products. What is not clear is the extent to which substitution leads to the WHO’s goal of reducing the overall levels of alcohol consumption in populations and consumer groups. For beer, British household purchase data suggested that such reductions would be minimal due to the very low levels of overall purchases of zero-alcohol beers (alcohol by volume, ABV, = 0.0%) and non-zero no-alcohol beers (ABV > 0.0% and ≤0.5%) [8], although reductions might be greater for reformulation of existing higher-strength beers to contain less alcohol [6].

In this paper, we examine the extent to which substitution might be associated with reductions in purchases of grams of alcohol at the household level in Spain. Since the 1970s, there have been large decreases in per capita levels of alcohol consumption in Spain, which is largely due to decreases in wine consumption and increases in beer consumption, such that Spain is now a predominantly beer-drinking culture [9]. Over the last ten years, consumption has remained relatively stable [9]. In Spain [7], purchases of zero and no-alcohol beers and wines are nearly six-fold higher than in Great Britain [8] albeit at relatively low levels. This paper, therefore, considers the extent to which changes over time (for the period 2nd quarter 2017 to end of 1st quarter 2022) in purchases of zero-alcohol (ABV = 0.0%) beers and wines, and changes in purchases of non-zero no-alcohol beers (ABV > 0.0% and ≤1.0%, the upper defined level of a no-alcohol beer in Spain) are associated with changes in the purchase of overall amounts of grams of alcohol, and the extent to which any changes might vary by sociodemographic attributes of the households.

If substitution, as proposed by the WHO [3], results in decreasing alcohol consumption, then it is important to consider policy options that might favour substitution. Analyses of British household purchase data have indicated that shifts from higher to lower-strength products can be facilitated by price, specifically by decreasing the price per gram of lower-strength products and increasing the price per gram of higher-strength products [10,11]. Modelling studies have also suggested that greater shifts could take place, and thus greater reductions in alcohol consumption, if price changes were not just related to grams of alcohol but also to ABV [12]. Specifically, in the United Kingdom, a fixed duty per gram of alcohol that doubled with an ABV of between 2.0% and 5.0% and then doubled again between and ABV of more than 5.0% and 40% would result in an additional reduction in overall alcohol consumption of 5.4% compared to the existing tax regime [12]. We therefore consider how changes in purchases relate to changes in price both for price per gram of alcohol and for price per gram of alcohol per ABV.

Thus, the two research questions that this paper addresses are: (i) is there a link between substitution and decreased alcohol consumption; and (ii) for beer, which has been the most studied product of substitution [4], how do changes in purchases of no-alcohol beers relate to changes in price both for price per gram of alcohol and for price per gram of alcohol per ABV?

The specific hypotheses to be tested are:Increases in purchases of zero-alcohol beers and wines and non-zero no-alcohol beers over time are associated with decreased purchases of grams of all alcohol;Decreases in the ABV of other purchased beers and wines over time are associated with decreased purchases of grams of all alcohol that are greater than the decreases associated with increases in purchases of zero-alcohol beers and wines and non-zero no-alcohol beers over time;Decreases in the price of zero- and no-alcohol beers (ABV ≤ 1.0%) and increases in the price of all other beers over time are associated with decreased purchases of grams of all alcohol; andIncreases in the price of all other beers per gram over time that are adjusted per ABV are associated with greater decreased purchases of grams of all alcohol than increases in the price of all other beers per gram, with no adjustment for ABV over time.

## 2. Methods

### 2.1. Study Design

We use ARIMA time-series analyses to investigate the potential impact of changes in purchases of zero- and no-alcohol products on changes in purchases of grams of all alcohol over time, investigating if this differs by sociodemographic attributes of households. We also use ARIMA time-series analyses to investigate the potential impact of changes in the price of beers over time on changes in purchases of grams of all alcohol over time

### 2.2. Data Source

Our data source is Kantar Worldpanel’s (KWP) household shopping panel. KWP comprises approximately 12,000 Spanish households at any one time, recruited via stratified sampling, with targets set for province, household size, and age of main shopper, with the panel being representative of households in Spain as a whole. Households provide demographic information when joining the panel, which is followed by annual updates and quality checks. Using barcode scanners, households record all alcohol purchases brought into the home from all store types, including Internet shopping.

We analysed raw KWP data on take-home purchasing of alcohol products in Spain for the time-period from the second quarter of 2017 to the end of the first quarter of 2022. For each individual purchase, the provided data included the type and volume of the purchase, the brand, the percent alcohol by volume (ABV), and the price paid (€). For one-third of purchases, a band of ABV was provided rather than the specific ABV; for these products, the mid-value of the ABV band was taken. The provided data categorised each purchase as either a beer, a wine, a sparkling wine, or a spirits-based product. The volume purchased was combined with ABV to calculate grams of alcohol purchased. For no-alcohol beers, we calculated the price per litre paid; and, for beers with an ABV > 3.5%, we calculated the price (€cents) per gram of alcohol paid, and the price (€cents) per gram of alcohol paid, adjusted for ABV (€/gram/ABV). Households were grouped by age of main shopper, social grade, and autonomous community; see Appendix A.

We prepared data by, first, for any day that a household bought alcohol, summing the amount of alcohol purchased in both volume and grams, divided by the number of adults in the household. Then, for each day of the time-period (second quarter of 2017 to end of first quarter of 2022), we calculated the mean volume and grams of purchases for all products and for beers and wines, the mean alcohol by volume of purchases for beers and wines, and the mean price paid for beers across all households. For beers and wines, we made separate calculations for: zero-alcohol (ABV0.0%) beers and wines; non-zero no-alcohol beers (ABV > 0.0% and ≤1.0%); no-alcohol beers, including zero-alcohol beers (ABV ≤ 1.0%), and all other beers and wines.

### 2.3. Statistical Analyses

We used generalised linear models to estimate the proportion of households that purchased zero- and no-alcohol products by the sociodemographic attributes of the households, estimating odds ratios by the sociodemographic attributes of the households. For details, see Appendix A. We plot graphs and tabulate the data of the proportions and odds ratios with 95% confidence intervals.

#### 2.3.1. Associations between Purchases of Zero- and No-Alcohol Beers and Wines and the ABV of Beers and Wines and Purchases of Grams of Alcohol

To assess the associations between purchases of zero- and no-alcohol beers and wines and the ABV of beers and wines and purchases of grams of alcohol, we used Auto-Regressive Integrated Moving Average (ARIMA) models with multiple independent variables. The dependent variable was the total number of grams of alcohol purchased per adult per household per day of purchase averaged across all households for each day from the second quarter of 2017 to the end of the first quarter of 2022. We ran two separate models with the following independent variables:

Model 1: Volume of purchases (mL) of zero-alcohol beer; volume of purchases (mL) of non-zero no-alcohol beer; volume of purchases (mL) of zero-alcohol wine; ABV of purchases of all other beer; and, ABV of purchases of all other wine;

Model 2: Same variables for Model 1 plus: volume of purchases (mL) of all other beer; volume of purchases (mL) of all other wine;

For details, see Appendix A. We report unstandardised coefficients with 95% confidence intervals.

We repeated Model 2 separately for each of the household characteristics (four groups of age of main shopper; four groups of level of affluence of the household; and 17 autonomous communities (regions) of Spain. We plot unstandardised coefficients with 95% confidence intervals of the associated changes over time between volume of purchases (ml) of zero-alcohol beer and the total number of grams of alcohol purchased.

#### 2.3.2. Associations between Price of No-Alcohol and Other Beers and Purchases of Grams of Alcohol

For prices of beer with an ABV > 3.5%, we contrast prices (€cent/gram and €cent/gram/ABV) against ABV, and prices (€cent/gram and €cent/gram/ABV) against number of grams of alcohol purchased within beer. Since the two price measures (€cent/gram and €cent/gram/ABV) are on different scales, we use standardised values of prices, ABV and grams. We undertake curve estimations of the relationships, taking as the model that curve estimation with the highest R^2^, which, in all cases, was a cubic relationship. We calculate the predicted values of the dependent variable based on the coefficients arising for the cubic model and plot the dependent variables against the independent variables. For details, see Appendix A.

To assess the associations between the price of no-alcohol and other beers and purchases of grams of alcohol, we used Auto-Regressive Integrated Moving Average (ARIMA) models with three dependent variables (grams of all alcohol purchased, ABV of all beer with an ABV > 3.5%, and volume (ml) of purchased no-alcohol beer), and three independent variables (price of no-alcohol beer (€/litre), and the two separate prices of all beer with an ABV > 3.5% (€cent/gram and €cent/gram/ABV). For details, see Appendix A. We report both standardised and unstandardised coefficients with 95% confidence intervals.

All analyses were performed with SPSSv27 [13].

## 3. Results

### 3.1. Households and Purchases

We analysed data from 18,954 Spanish households with 1.29 million separate alcohol purchases between the beginning of the second quarter of 2017 to the end of the first quarter of 2022. Over the full time span, the total number of grams of alcohol purchased per adult per household per day of purchase, averaged over all study days (from the second quarter of 2017 to the end of the first quarter of 2022) was 96.7 g (95% CI = 96.5 to 96.9). Out of the volume of all beers purchased (per adult per household per day of purchase per study day) (1187 mL, 95%CI = 1179 to 1195), 8.8% (95% CI = 8.7 to 9.0) was from zero-alcohol beers and 7.3% (95%CI = 7.2 to 7.4) was from non-zero no-alcohol beers. Out of the volume of all wines purchased (446 mL, 95%CI = 444 to 449 mL), 6.4% (95% CI = 6.2 to 6.6) was from zero-alcohol wines.

The proportions of all households that had made at least one separate purchase of zero-alcohol beer, non-zero no-alcohol beer and zero-alcohol wine are plotted by autonomous community (region of Spain) in which the household is located (ordered from community with highest GDP per capita to lowest), age group of the main household shopper, and social grade (affluence) of household in Appendix A, with the proportions and odds ratios for purchase documented in Appendix A. By age group, the proportions of households buying zero- or no-alcohol products tended to be greater the higher the age of the main shopper. There was no discernible pattern by autonomous community, except for zero-alcohol wine, in which the three autonomous communities with the lowest GDP per capita tended to have the highest proportions. By social grade, there was no discernible pattern, except for non-zero no-alcohol beer, for which the proportions tended to be higher the lower the social grade.

### 3.2. Associations between Purchases of Zero- and No-Alcohol Beers and Wines and the ABV of Beers and Wines and Purchases of Grams of Alcohol

Figure 1 plots the trends over time for the primary dependent variable (total grams of alcohol purchased) and the independent variables from Model 1. Over time, there were slight increases in purchases of grams of alcohol and volume of zero-alcohol beer, and slight decreases in purchases of non-zero no-alcohol beer and zero-alcohol wine; see Appendix A. During 2020, there was an upsurge in total grams of alcohol purchased, which was likely due to COVID-19 lockdowns, a phenomenon observed elsewhere [14]. The ABV of all other beer and of all other wine increased very slightly over time.

At the level of the household, for every 100 mL increase in purchases of zero-alcohol beer, associated purchases of grams of all alcohol dropped by 1.4 g (95%CI = 0.2 to 2.6), a 1.5% drop (Model 1), which when adding volume of purchases of all other beers and wines (Model 2), decreased to 0.8 g (95%CI = 0.3 to 1.4); see Table 1. For every 100 mL increase in purchases of non-zero no-alcohol beer, associated purchases of grams of all alcohol dropped by 2.4 g (95%CI = 1.0 to 3.8), a 2.6% drop (Model 1), which when adding volume of purchases of all other beers and wines (Model 2), decreased to 0.8 g (95%CI = 0.1 to 1.5). Changes in purchases of zero-alcohol wine were not associated with changes in purchases of all grams of alcohol. Changes in the alcohol by volume of all other beers and wines had a marked impact. For every decrease in the absolute ABV of beer of 1.0% (e.g., from a 5.0% beer to a 4.0% beer), the associated reductions in purchases of all grams of alcohol was 13.2 (95%CI = 7.6 to 18.0), Model 1, a 14.3% drop; for wine, the drop was 7.4 g (95% CI = 5.1 to 9.7), an 8.0% drop.

Appendix A, plots the ARIMA coefficients for the associations between changes in purchase of zero-alcohol beer and changes in purchase of all grams of alcohol by each of the sociodemographic attributes of the household. By age group, the largest associated drop was for the 35–49-year-olds. By social grade and affluence, there was an indication that households located in autonomous communities with higher GDP per capita and households with higher social grades had larger associated drops than households located in autonomous communities with lower GDP per capita and households with lower social grades. A regression equation found that for every €1000 increase in GDP per capita, the associated decrease in grams of alcohol purchased for every increase in one millilitre of zero-alcohol beer purchased was larger by 8.7^−4^ g (95%CI = 3.0^−4^ to 1.4^−3^).

### 3.3. Associations between Price of No-Alcohol and Other Beers with Purchases of Grams of Alcohol

At the level of each purchase, Appendix A, demonstrate that the price per gram of alcohol within beer that had an ABV >3.5% did not increase with increasing strength of ABV; and for every increase in price per gram, at least up to the standardised value of price of 0.0, the number of grams of alcohol purchased within beer was less when the price was expressed as €cent/g/ABV than when it was expressed as €cent/g.

Figure 2 plots the changes in price per one litre of no-alcohol beer (ABV ≤ 1.0%), per one gram of all other beer (ABV > 3.5%), and per one gram per ABV of all other beer (ABV > 3.5%) over time (the independent variables), together with the three dependent variables, grams of all alcohol (expressed as hectograms), ABV of all other beer with an ABV > 3.5% (divided by 10 to fit on vertical axis), and volume of purchases of no-alcohol beer (expressed as decilitres). All variables increased slightly over time; see Appendix A.

Table 2 displays the ARIMA regression coefficients between changes in price and changes in purchases over time. Comparing the standardised coefficients (that places the coefficients on an equivalent scale of standard deviations) shows that increases in the price of beers with an ABV > 3.5%, per gram of alcohol, adjusted for the ABV, are associated with greater increases in purchases of no-alcohol beer, ABV ≤ 1.0% (coefficient for one standard deviation change in price = 0.35, 95%CI 0.12 to 0.58), than decreases in the prices of no-alcohol beer (coefficient for one standard deviation change in price 0.08, 95%CI 0.03 to 1.26). Likewise, increases in the price of beers with an ABV > 3.5%, per gram of alcohol, adjusted for the ABV, are associated with greater decreases in purchases of all grams of alcohol (coefficient for one standard deviation change in price 0.33, 95%CI 0.09 to 0.56) than increases in prices of beers with an ABV > 3.5%, per gram of alcohol, not adjusted for the ABV, which actually show an associated small increase in purchases of grams of all alcohol (coefficient for one standard deviation change in price 0.056, 95%CI 0.005 to 0.107).

## 4. Discussion

### 4.1. Main Findings

Our four hypotheses are confirmed. For hypothesis one (increases in purchases of zero-alcohol beers and wines and non-zero no-alcohol beers over time are associated with, albeit very small, decreased purchases of grams of all alcohol), there were associations between increases in household purchases of zero- and non-zero- no-alcohol beers (but not zero-alcohol wines) over time and decreases in purchases of grams of all alcohol over time. For example (Model 2, Table 1), when they made an alcohol purchase, if all households increased their purchases of zero-alcohol beer by 100 mL, associated purchases of grams of all alcohol would drop by 0.83% (95%CI = 0.31 to 1.45).

For hypothesis two (decreases in the ABV of other beers and wines over time are associated with decreased purchases of grams of all alcohol that are greater than the decreases associated with increases in in purchases of zero-alcohol beers and wines and non-zero no-alcohol beers over time), at the household level, there were large associations between decreases in the purchased ABV of both higher-strength beers and wines and decreases in purchases of grams of all alcohol. For example (Model 2, Table 1), for every 1% decrease in the absolute ABV of wine (for all non-zero ABV wines), associated purchases of grams of all alcohol would drop by 5.7% (95%CI = 4.5 to 6.8). This suggests that the reformulation of existing beers and wines to contain less alcohol could have a much greater impact on reducing purchases of grams of all alcohol than relatively small increases in purchases of zero- and no-alcohol beers and wines.

For hypothesis three (decreases in the price of no-alcohol beers (ABV ≤ 1.0%) and increases in the price of all other beers (ABV > 3.5%) over time are associated with decreased purchases of grams of all alcohol) and four (increases in the price of all other beers (ABV > 3.5%) per gram over time that increase with increasing ABV are associated with greater decreased purchases of grams of all alcohol than increases in the price of all other beers per gram, with no adjustment for ABV over time), the changes that made the greater impact were changes in the price of grams of alcohol within beers with an ABV > 3.5% adjusted for the ABV of the beer. Increases in this adjusted price had a greater association with increasing purchases of no-alcohol beers and decreasing purchases of grams of all alcohol than deceases in the price of no-alcohol beer and increases in the price of beers with an ABV > 3.5% that was not adjusted for ABV. For example (Table 2), a one standard deviation in the price per gram of beer (ABV > 3.5%), adjusted for the ABV of the beer was associated with a four-times greater one standard deviation increase in the volume of purchases of no-alcohol beer (ABV ≤ 1.0%) than a one standard deviation decrease in the price per litre of no-alcohol beer.

### 4.2. What Is Already Known on This Topic

The only other similar study that we are aware of is our own study of British household purchase data with over four million alcohol purchases from 69,803 households for the years 2015–2019, in which households were clustered as predominantly beer-purchasing, wine-purchasing, or spirits-purchasing. For predominantly beer-purchasing households, the analyses found that for every 10 mL increase in purchases of zero-alcohol beer (ABV = 0.0%) per adult per household per day, purchases of grams of all alcohol contained within beer dropped by 1.1% [8], for every drop in the absolute value of the ABV of beer of 0.1% over time (from a baseline of 4.34), purchases of grams of all alcohol contained within beer dropped by 6.9% [8], and, for predominantly wine-purchasing households, for every 5 mL increase in purchases of no-alcohol wine products per adult per household per day, purchases of grams of all alcohol contained within wine dropped by 1.2% [8].

### 4.3. What This Study Adds

A strength of the study is that we include a large number of alcohol purchases from a large number of households, with large numbers of data points before and after the examined events, with scanned barcode data providing objective data. Examining household purchases in Spain, with a very different drinking culture than that of Great Britain, found evidence for substitution and reduced purchases of grams of alcohol not only for the impact of no-alcohol beers and wines but also for the impact of reduced alcoholic strength variants of spirits products. Examined for beers and wines, the associated reductions occurred across all age and social grade groups, and, when examined by an autonomous community, were higher the higher the initial volume of purchases.

### 4.4. Limitations of the Study

A main limitation of the study is that we are only able to assess changes in household alcohol purchases from shops and supermarkets, and we exclude alcohol consumed from cafés, bars and restaurants. Furthermore, we only examine purchases and not actual levels of alcohol consumption for the time periods studied. Adults in a household may not have an equal share of the alcohol purchased, and not all adults in a household may be drinkers. The data also have limitations, with heavy drinkers tending to be under-represented in household panel data [14] and with alcohol purchases tending to be under-reported in these datasets [15,16].

## 5. Conclusions

From the perspective of household purchases, at least within both Great Britain and, with the present findings, Spain, two conclusions are prominent. First, the product reformulation of existing beers and wines to contain less alcohol has the potential to have a much greater impact in reducing grams of alcohol purchased (and, presumed, consumed) than increases in the current relatively low levels of purchases of no-alcohol beers and wines, notwithstanding that such increases are important. Second, increasing the prices of higher-strength beers, with prices per gram adjusted for the alcoholic strength of the beer, is key to both stimulating purchases of no-alcohol beers and decreasing purchases of grams of alcohol overall.

## Figures and Tables

**Figure 1 nutrients-14-04209-f001:**
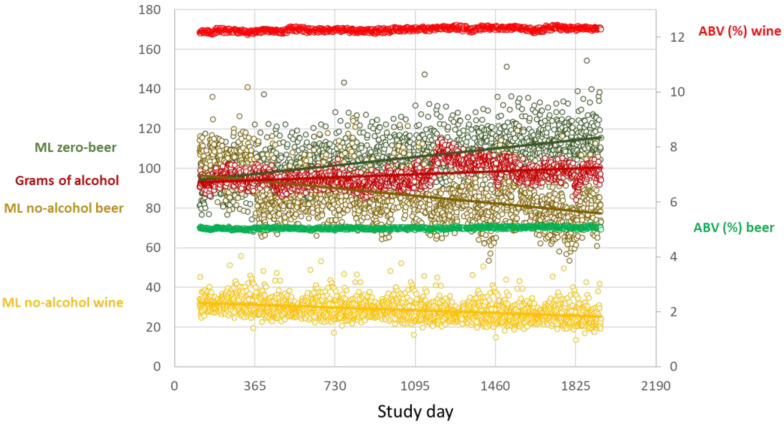
Left axis: plots of purchases per adult per household per day of purchase, averaged across all households for each study day (Day 114 = 1st April 2017; Day 1933 = 31st March 2022) for: grams of all alcohol, ml of zero-alcohol beer; ml of non-zero no-alcohol beer; and ml of zero-alcohol wine. Right axis: mean alcohol by volume (ABV) per purchase per study day across all households for all wine with an ABV > 0.0% and for all beer with an ABV > 1.0%. Each circle: daily data point. Straight lines: regression lines.

**Figure 2 nutrients-14-04209-f002:**
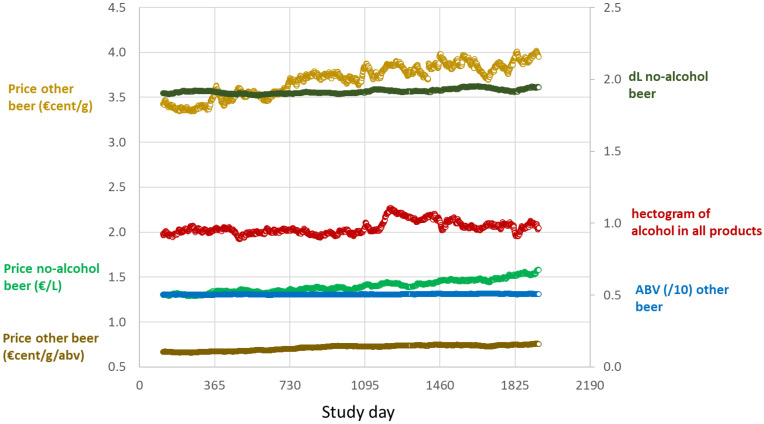
Left axis: plots of price of no-alcohol beer (€/litre), and price of all other beer (€/litre and €/litre/ABV) per purchase per day of purchase, averaged across all households for each study day (Day 114 = 1st April 2017; Day 1933 = 31st March 2022) Right axis: volume of no-alcohol beer (dL), grams of alcohol in all products (hectograms), and ABV (divided by ten to fit on vertical scale) of all other beer (with ABV > 3.5%) per purchase day per study day across all households. Each circle: daily data point.

**Table 1 nutrients-14-04209-t001:** ARIMA regression coefficients for associations between changes over time of listed independent variables and changes over time of dependent variable, grams of all alcohol purchased. All variables: per adult per household per day of purchase, averaged over each study day from day 114 to day 1933.

Model 1		Model 2	
Unit change 1 mL of zero-alcohol beer (ABV = 0.0%)	−0.014 (−0.026 to −0.002)	Unit change 1 mL of zero-alcohol beer (ABV = 0.0%)	−0.008 (−0.014 to −0.003)
Unit change 1 mL of non-zero no-alcohol beer (ABV ≥ 0.0% and ≤1.0%)	−0.024 (−0.038 to −0.010)	Unit change 1 mL of non-zero no-alcohol beer (ABV ≥ 0.0% and ≤1.0%)	−0.008 (−0.015 to −0.001)
Unit change 1 mL of zero-alcohol wine (ABV = 0.0%)	Non-significant	Unit change 1 mL of zero-alcohol wine (ABV = 0.0%)	Non-significant
Unit change 1% ABV of beer (ABV > 1.0%)	13.178 (7.643 to 18.712)	Unit change 1% ABV of beer (ABV > 1.0%)	9.535 (6.887 to 12.183)
Unit change 1% ABV of wine (ABV > 0.0%)	7.398 (5.126 to 9.670)	Unit change 1% ABV of wine (ABV > 0.0%)	5.496 (4.375 to 6.618)
		Unit change 1 mL of higher-strength beer (ABV ≥ 1.0%)	0.104 (0.100 to 0.107)
		Unit change 1 mL of higher-strength wine (ABV ≥ 1.0%)	0.040 (0.038 to 0.042)

**Table 2 nutrients-14-04209-t002:** ARIMA regression coefficients for associations between changes over time of listed independent variables and changes over time of dependent variables. For each dependent variable, all independent variables added to model. All variables: per adult per household per day of purchase, averaged over each study day from day 114 to day 1933.

	Dependent Variables					
Independent Variables	Volume of No-Alcohol Beer (ABV ≤ 1.0%) (mL)		ABV (%) of All Other Beer (For Beer with ABV > 3.5%)		Grams of Alcohol in All Products	
	Unstandardised	Standardised	Unstandardised	Standardised	Unstandardised	Standardised
Mean across time span	191.6 (189.1 to 194.10	0.0	5.051 (5.047 to 5.056)	0.0	96.6 (96.2 to 97.1)	0.0
Decrease in price of no-alcohol beer (ABV ≤ 1.0%) (€/L)	36.6 (13.2 to 60.1)	0.08 (0.03 to 1.26)	Non-significant	Non-significant	−5.85 (−10.49 to −1.20)	−0.06 (−0.11 to −0.01)
Increase in price of all other beer with ABV > 3.5% (€cent/g)	−7.60 (−12.30 to −2.90)	−0.39 (−0.63 to −0.15)	1.05 (0.99 to 1.11)	3.02 (2.85 to 3.20)	2.15 (0.19 to 4.11)	0.056 (0.005 to 0.107)
Increase in price of all other beer with ABV > 3.5% (€cent/g/0.1ABV%)	35.7 (12.6 to 58.8)	0.35 (0.12 to 0.58)	−0.51 (−0.54 to −0.48)	−2.835 (−2.999 to −2.66)	−6.46 (−11.09 to −1.83)	−0.33 (−0.56 to −0.09)

## Data Availability

No additional data are available. Kantar WorldPanel data cannot be shared due to licensing restrictions.

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
