# Peer review of "Effects of Substitution of Higher-Alcohol Products with Lower-Alcohol Products on Population-Level Alcohol Purchases: ARIMA Analyses of Spanish Household Data"

_nutrients, 2022, doi:10.3390/nu14194209_

Round 1

Reviewer 1 Report

No comments.

Reviewer 2 Report

Overall, the study has many strengths including a robust sample size and sampling strategy. The authors concluded that reducing the strength of higher alcohol beers and increasing price decreased consumption of total alcohol grams and reduced purchasing of the higher ABV beers. It’s difficult to tell from the study design whether the change in alcohol purchasing behavior is influenced by other factors like personal preference or are heavily influenced by the price elasticity of demand. The study timeframe overlaps the COVID lockdown and it’s unclear the full extent the pandemic had on alcohol consumption. I think the main point of the study is the price influence on the strength of beer people purchase rather than changing the strength of the beer alone.